# Disproportionality analysis of drug-associated progressive multifocal leukoencephalopathy using spontaneous reports: A 20-year signal detection study based on the FAERS database

**Hailong Wang, Xinyi Li, Lijuan Shangguan** *

Shanxi Bethune Hospital, Shanxi Academy of Medical Sciences, Third Hospital of Shanxi Medical University, Tongji Shanxi Hospital, Taiyuan, China

* shangguanlijuan@sxbqeh.com.cn

## Abstract

### Background

Progressive multifocal leukoencephalopathy (PML) is a rare, often fatal demyelinating disease caused by JC virus reactivation in immunocompromised patients. With the increasing use of immunosuppressants and biologics, PML reporting in non-HIV populations is rising. This study aimed to evaluate drug-associated PML reporting signals using real-world pharmacovigilance data.

### Methods

We analyzed FAERS database from 2004Q1 to 2024Q4. We identified PML reports via MedDRA Terms and manual validation. Four algorithms **(ROR, PRR, BCPNN, MGPS)** were jointly applied, with drugs showing signals across all four defined as high-risk.

### Results

7,244 PML reports involving 298 drugs were identified; 72 drugs showed consistent signals, predominantly immunomodulators (e.g., natalizumab, rituximab), antineoplastics, and biologics. High-risk indications included multiple sclerosis, lymphoma, autoimmune diseases, and organ transplantation. PML reporting increased substantially in non-HIV populations. Time-to -reporting varied widely (49–1343days). Over one-third of reports were associated with life-threatening outcomes or death.

### Conclusions

This analysis identified 72 drugs with consistent PML reporting signals. However, these findings represent statistical associations in spontaneous reports, not causal relationships or true incidence rates. Inherent limitations—including underreporting,

**Data availability statement:** The data analyzed in this study are publicly available from the US FDA Adverse Event Reporting System (FAERS; https://www.fda.gov/drugs/questions-and-answers-fdas-adverse-event-reporting-system-faers/fda-adverse-event-reporting-system-faers-public-dashboard). All data generated or analyzed during this study are included in this published article and its supplementary information files.

**Funding:** This work was supported by the Shanxi Provincial Department of Science and Technology, Basic Research Program (Free Exploration Category) — Youth Project [Grant No. 202403021222401, amount: CNY 50,000, received by SG LJ (Shang Guan Lijuan)]. Funder website: http://kjt.shanxi.gov.cn/. The sponsors/funders had no role in the study design, data collection and analysis, decision to publish, or preparation of the manuscript.

**Competing interests:** The authors have declared that no competing interests exist.

incomplete medication histories, and lack of exposure denominators—require cautious interpretation. Prospective validation studies are essential to establish causality and quantify absolute risks.

## Introduction

Progressive multifocal leukoencephalopathy (PML) is a rare and devastating central nervous system disease caused by reactivation of the JC virus (JCV) in immunosuppressed individuals. It typically presents with subacute onset of cognitive impairment, motor dysfunction, language disturbances, and visual field deficits, progressing rapidly with poor prognosis [1]. The diagnosis of PML relies on a combination of clinical manifestations, magnetic resonance imaging (MRI), and detection of JCV DNA in cerebrospinal fluid [2].

During the 1980s and 1990s, PML was primarily observed in patients with HIV/AIDS and represented a significant opportunistic infection in this population [3,4]. Since the introduction of highly active antiretroviral therapy (HAART), the incidence of HIV-associated PML has declined significantly [5,6]. However, in recent years, PML cases have continued to rise in non-HIV populations [7–9]. This increase coincides with the growing use of immunosuppressants and biologic agents for conditions like multiple sclerosis, autoimmune diseases, and solid tumors [10,11]. HIV-associated and non-HIV-associated PML differ in several key aspects. HIV-associated PML often presents with inflammatory features on MRI, particularly in the context of immune reconstitution inflammatory syndrome (IRIS), and generally responds to immune restoration through HAART, with median survival extending to 1.8–4.3 years after diagnosis, as reported by some studies, with some cohorts approaching or exceeding 4 years [8,9]. In contrast, non-HIV PML typically shows poorer outcomes, with median survival of only 2–6 months in many patient groups, particularly in those with hematological malignancies [8].

Nevertheless, PML remains an exceptionally rare condition, with very low incidence in the general population and only a few cases per thousand in high-risk groups like natalizumab-treated multiple sclerosis patients [12–14]. Due to this rarity, comprehensive studies on potential drug-PML relationships are limited in real-world settings. Current evidence largely comes from case reports [15] or small retrospective analyses [16,17]. Many studies suffer from short follow-up periods, inconsistent diagnostic criteria, and challenges in establishing definitive causal relationships between specific medications and PML development.

The US Food and Drug Administration Adverse Event Reporting System (FAERS) database contains over 15 million reports since 1968. It is critical to recognize that FAERS reflects adverse event reporting patterns rather than true disease incidence or prevalence in the population. Signal detection methods, including disproportionality analyses, identify statistical associations that suggest potential safety concerns but do not establish causal relationships between drugs and adverse events [18–20].

This study builds on prior knowledge using FAERS data spanning 2004–2024 (updated to 2024Q4), we provide an all-drug, indication-agnostic panoramic screen

of drug–PML reporting. To improve specificity, we apply four complementary disproportionality algorithms—Reporting Odds Ratio (ROR), Proportional Reporting Ratio (PRR), Bayesian Confidence Propagation Neural Network (BCPNN), and Multi-item Gamma Poisson Shrinker (MGPS)—and define a signal only when all four methods are concordant, yielding an updated and conservative signal inventory to prioritize drugs for further validation.

## Materials and methods

### 1. Data source and report identification

This study utilized data from the US Food and Drug Administration Adverse Event Reporting System (FAERS) database [21] (https://www.fda.gov/drugs/drug-approvals-and-databases/fda-adverse-event-reporting-system-faers-database), including all spontaneously reported adverse drug event records worldwide from the first quarter of 2004 to fourth quarter of 2024. The FAERS database contains multiple tables, including demographic (DEMO), drug (DRUG), indication (INDI), adverse reaction (REAC), and outcome (OUTC) information.

Reports of PML were identified using the MedDRA Preferred Term "Progressive multifocal leukoencephalopathy." Identified reports underwent validation through review of narrative fields and supporting documentation by two independent reviewers, with disagreements resolved by a third reviewer. Reports with incomplete information or where PML could not be confirmed were excluded from the analysis.

### 2. Drug and indication classification

All drugs were standardized to International Nonproprietary Names (INN), taking into account possible synonyms and spelling variations. Vaccines, diagnostic agents, and combination preparations were excluded. Herbal medicines and records with unclear drug indications were also excluded. Indications were categorized based on the MedDRA classification system into major underlying disease groups: autoimmune diseases (including multiple sclerosis, rheumatoid arthritis, and inflammatory bowel disease), malignancies (hematological and solid tumors), organ transplantation, and other conditions. For reports with multiple indications, the primary underlying disease was determined based on the reported indication for the suspected causative drug, clinical context, and medical literature.

### 3. Signal detection methods

Four mainstream pharmacovigilance signal detection methods were jointly applied, including two frequentist disproportionality measures (ROR [22,23] and PRR [24,25]) and two Bayesian shrinkage approaches (BCPNN [18,26] and MGPS [19,26]). ROR/PRR are derived from $2 \times 2$ contingency tables and are intuitive to interpret but can be sensitive to sparse counts, whereas BCPNN/MGPS use Bayesian shrinkage to stabilize estimates for rare drug–event pairs. We therefore required concordant signals across all four methods to prioritize specificity and reduce spurious findings due to random variation, acknowledging a potential loss of sensitivity [27].

- ROR: $ROR > 1$ and 95% confidence interval (CI) lower bound $> 1$;
- PRR: $PRR > 2$ and $\chi^2 > 4$ and $N \geq 3$;
- BCPNN: Information Component (IC025) $> 0$;
- MGPS: $EBGM05 \geq 2$.

### 4. Ethics statement

This study used publicly available, anonymized data from the FAERS database and did not involve direct patient intervention or identifiable personal information. This research qualified for exemption from ethics committee approval, consistent

with previous pharmacovigilance studies using FAERS data [28]. Data access and analysis complied with the FDA's FAERS data use policies and applicable privacy regulations.

## Results

### 1. Data screening and reports summary

From the FAERS database, a total of 18,613,992 unique patient records were extracted from the first quarter of 2004 to the fourth quarter of 2024 after data deduplication and multi-table integration. Further screening resulted in 7,244 drug-PML pairings, involving 298 drugs potentially associated with PML. By applying four signal detection algorithms (ROR, PRR, BCPNN, and MGPS), the top 30 drugs ranked by the number of reports are presented in Table 1, with the complete list provided in S1 Table. The complete data processing and screening flow is presented in Fig 1.

### 2. Characteristics of reported PML reports in FAERS

A total of 7,244 PML reports were included in this study. The overall mean age of patients was 53.8 ± 15.6 years, with a mean body weight of 70.4 ± 18.8 kg. The overall male-to-female ratio was approximately 1:1.3, with females slightly predominant in reports. Subgroup analysis revealed that reports associated with HIV/AIDS were predominantly male, whereas females were notably higher in non-HIV/AIDS-associated reports (Fig 2e).

Reporting of PML to FAERS has increased substantially in recent years, while HIV/AIDS-related reports remained stable at 20–30 per year. Non-HIV/AIDS reports now account for several hundred annually over the past decade (Fig 2a). These trends reflect reporting patterns and drug utilization changes rather than disease incidence. More than a third of reported cases documented life-threatening complications or death (Fig 2b). Geographically, Europe and North America contributed the most reports (Fig 2c). Physicians and healthcare professionals were the primary reporters of these reports (Fig 2d). These patterns reflect reporting behaviors and healthcare system characteristics rather than true PML epidemiology.

### 3. Results of drug signal detection

Our pharmacovigilance analysis identified 72 drugs showing disproportionate PML reporting patterns using four independent signal detection algorithms (Table 1 and S1 Table). Fig 3 presents the forest plot of reporting odds ratios (RORs) with 95% confidence intervals for these agents, arranged in descending order based on the report frequency.

Among the signal-positive drugs, natalizumab and rituximab had the highest reporting volumes (1,848 and 1,298 reports, respectively) with corresponding ROR values of 38.08 (95% CI: 36.11–40.15) and 39.44 (95% CI: 37.13–41.9). These were followed by mycophenolic acid (271 reports; ROR: 12.13, 95% CI: 10.74–13.7) and fingolimod (249 reports; ROR: 8.01, 95% CI: 7.06–9.09). The plot demonstrates substantial heterogeneity in reporting associations across drugs, with ROR values ranging from 2.91 (95% CI: 2.04–4.17) for ofatumumab to 40.42 (95% CI: 24.65–66.26) for chlorambucil.

### 4. Analysis of drug indications and disease classification

**4.1. Pharmacological classification of PML-related drugs.** As shown in Fig 4, we classified the 72 PML-associated drugs into five major pharmacological categories. The analysis revealed that immunomodulatory drugs represented the largest proportion, which were further divided into three subcategories: drugs targeting lymphoid B and T cells (3,443 reports, approximately 60% of all reports), broad-spectrum immunosuppressants (551 reports), and small molecule immunomodulators (468 reports). The other major categories included hormone drugs (424 reports), cytotoxic drugs (728 reports), therapeutic drugs (103 reports), and antiviral agents (279 reports). The pie chart clearly illustrates the distribution of these drug categories, with the blue section representing the major category of immunomodulatory drugs, which includes the three subcategories mentioned above. Collectively, these three subcategories account for approximately 77% of all PML-related reports.

**Table 1. Top 30 (of 72) drugs showing positive PML signals in all four detection algorithms, ranked by report numbers. (complete list provided in the S1 Table).**

| DRUG | a | ROR (95% CI) | PRR (χ²) | EBGM(EBGM05) | IC(IC025) |
|---|---|---|---|---|---|
| NATALIZUMAB | 1848 | 38.08 (36.11–40.15) | 37.67 (49155.73) | 28.31 (27.08) | 4.82 (4.75) |
| RITUXIMAB | 1298 | 39.44 (37.13–41.9) | 38.96 (39423.66) | 32.16 (30.58) | 5.01 (4.92) |
| MYCOPHENOLIC ACID | 271 | 12.13 (10.74–13.7) | 12.08 (2652.3) | 11.67 (10.54) | 3.54 (3.37) |
| FINGOLIMOD | 249 | 8.01 (7.06–9.09) | 7.99 (1470.99) | 7.75 (6.97) | 2.95 (2.77) |
| METHOTREXATE | 186 | 3.76 (3.25–4.35) | 3.75 (366.28) | 3.68 (3.26) | 1.88 (1.67) |
| PREDNISOLONE | 145 | 11.38 (9.65–13.42) | 11.34 (1339.95) | 11.13 (9.7) | 3.48 (3.23) |
| DOXORUBICIN | 134 | 10.82 (9.11–12.84) | 10.77 (1166.72) | 10.59 (9.18) | 3.41 (3.15) |
| TACROLIMUS | 133 | 5.83 (4.91–6.92) | 5.82 (520.97) | 5.73 (4.96) | 2.52 (2.27) |
| CYCLOPHOSPHAMIDE | 120 | 10.53 (8.79–12.62) | 10.49 (1013.64) | 10.33 (8.88) | 3.37 (3.1) |
| PREDNISONE | 113 | 8.37 (6.95–10.08) | 8.35 (719.72) | 8.23 (7.05) | 3.04 (2.77) |
| BENDAMUSTINE | 101 | 28.03 (23.01–34.15) | 27.74 (2568.13) | 27.37 (23.2) | 4.77 (4.49) |
| CICLOSPORIN | 89 | 4.64 (3.77–5.72) | 4.64 (250.82) | 4.59 (3.85) | 2.2 (1.89) |
| DEXAMETHASONE | 89 | 5.51 (4.47–6.8) | 5.5 (324.01) | 5.45 (4.57) | 2.45 (2.14) |
| FLUDARABINE | 84 | 27.63 (22.26–34.3) | 27.35 (2108.87) | 27.05 (22.57) | 4.76 (4.44) |
| METHYLPREDNISOLONE | 77 | 7.96 (6.36–9.97) | 7.94 (462.31) | 7.87 (6.52) | 2.98 (2.65) |
| OCRELIZUMAB | 57 | 2.91 (2.24–3.78) | 2.91 (70.84) | 2.89 (2.33) | 1.53 (1.15) |
| BORTEZOMIB | 51 | 4.08 (3.1–5.37) | 4.07 (117.54) | 4.05 (3.22) | 2.02 (1.62) |
| HYDROXYCHLOROQUINE | 51 | 8.19 (6.22–10.79) | 8.17 (318.75) | 8.12 (6.45) | 3.02 (2.62) |
| MIRTAZAPINE | 49 | 6.83 (5.16–9.05) | 6.82 (241.76) | 6.78 (5.36) | 2.76 (2.35) |
| BRENTUXIMAB VEDOTIN | 47 | 17.06 (12.8–22.75) | 16.96 (701.43) | 16.85 (13.25) | 4.07 (3.66) |
| ALEMTUZUMAB | 44 | 10.12 (7.52–13.62) | 10.08 (358.04) | 10.03 (7.82) | 3.33 (2.89) |
| OBINUTUZUMAB | 42 | 14.06 (10.37–19.06) | 13.99 (503.84) | 13.91 (10.79) | 3.8 (3.36) |
| LAMIVUDINE | 37 | 9.18 (6.64–12.69) | 9.15 (267.3) | 9.11 (6.95) | 3.19 (2.72) |
| BUSULFAN | 35 | 14.67 (10.52–20.47) | 14.59 (441.25) | 14.53 (11.53 (11) | 3.86 (3.38) |
| DARATUMUMAB | 33 | 7.16 (5.09–10.09) | 7.15 (173.71) | 7.12 (5.34) | 2.83 (2.33) |
| AZATHIOPRINE | 32 | 13.96 (9.86–19.78) | 13.9 (381.4) | 13.84 (10.34) | 3.79 (3.29) |
| LOPINAVIR; RITONAVIR | 31 | 12.73 (8.94–18.14) | 12.68 (332.1) | 12.63 (9.39) | 3.66 (3.15) |
| OFATUMUMAB | 30 | 2.91 (2.04–4.17) | 2.91 (37.52) | 2.9 (2.15) | 1.54 (1.02) |
| RITONAVIR | 29 | 11.23 (7.79–16.19) | 11.19 (268.02) | 11.15 (8.21) | 3.48 (2.95) |
| ETOPOSIDE | 27 | 5.87 (02–8.57) | 5.86 (43) | 5.84 (26) | 2.55 (2) |

Abbreviations: ROR (95% CI), Reporting Odds Ratio (95% Confidence Interval); PRR (χ²), Proportional Reporting Ratio (Chi-square); EBGM (EBGM05), Empirical Bayes Geometric Mean (5th percentile of posterior distribution); IC (IC025), Information Component (2.5th percentile of posterior distribution).

**4.2. Analysis of the correlation between drugs and diseases.** S1 Fig shows a heatmap of reporting associations between the top 30 drugs and 30 major diseases. Higher values in the heatmap represent stronger reporting associations between specific drugs and diseases. The heatmap demonstrates that natalizumab use in multiple sclerosis patients had the strongest reporting association (358 reports), followed by rituximab in non-Hodgkin's lymphoma (142 reports) and rituximab in B-cell lymphoma patients (142 reports).

**4.3. Classification and ranking of diseases related to PML.** S2 Fig presents the 17 major disease categories associated with PML after consolidating the top 30 diseases. Multiple sclerosis ranks first with 2,174 reports, followed by lymphoma (823 reports), leukemia (488 reports), plasma cell myeloma (307 reports), and HIV infection (289 reports). These five diseases represent the major proportion of all PML-related reports.

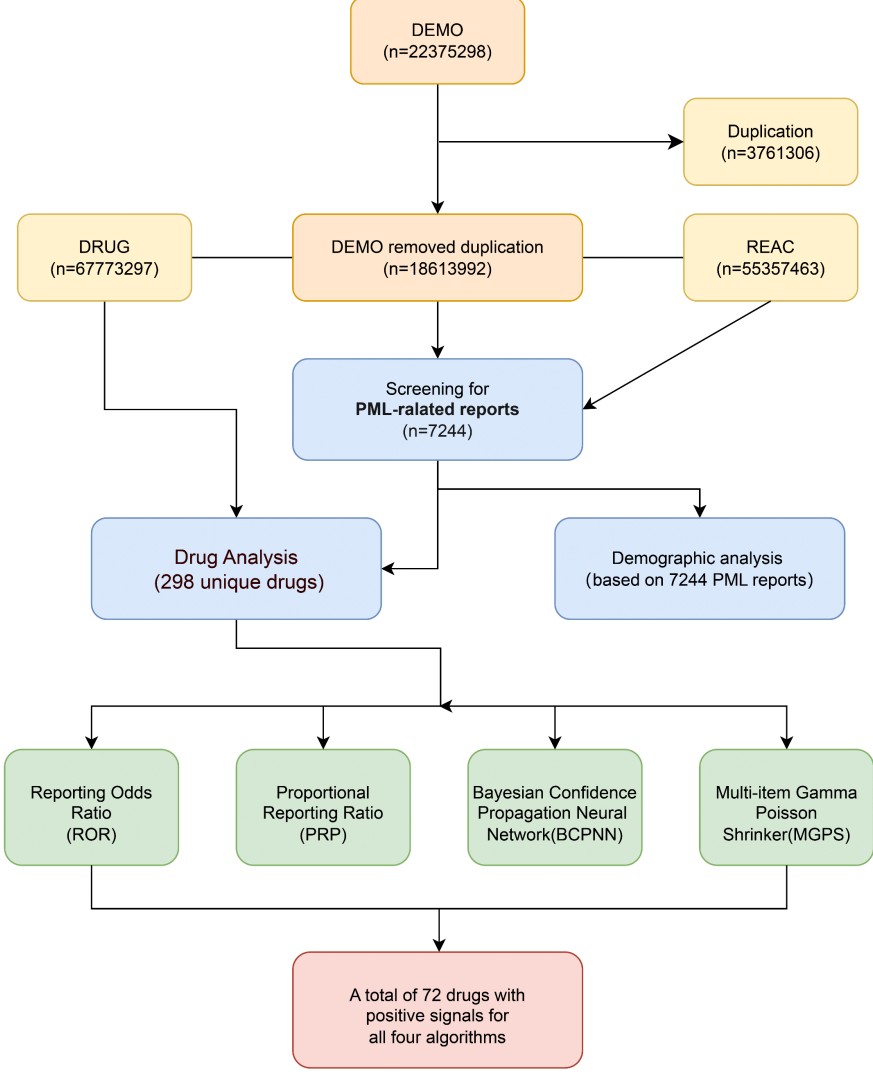

**Fig 1. Drug signal detection process for progressive multifocal leukoencephalopathy in FAERS.**

### 5. Onset timing and risk characteristics of major drugs

Among the 72 signal-positive drugs, the top 10 were selected for analysis of time-to-onset (TTO) from drug initiation to PML reporting. Median TTO varied widely across drugs, ranging from 49 days (ocrelizumab) to 1343 days (dimethyl fumarate). Ocrelizumab had the shortest median TTO (49 days), followed by natalizumab (262 days) and rituximab (335 days), while other drugs showed incrementally longer intervals (**Fig 5**).

## Discussion

This study systematically analyzed, based on the FAERS database, the epidemiological characteristics, drug signals, and clinical risk factors of drug-associated progressive multifocal leukoencephalopathy (PML) from 2004 to 2024.The major findings include: (1) Over the past two decades, the epidemiology of PML reports has shifted from HIV-related to non-HIV

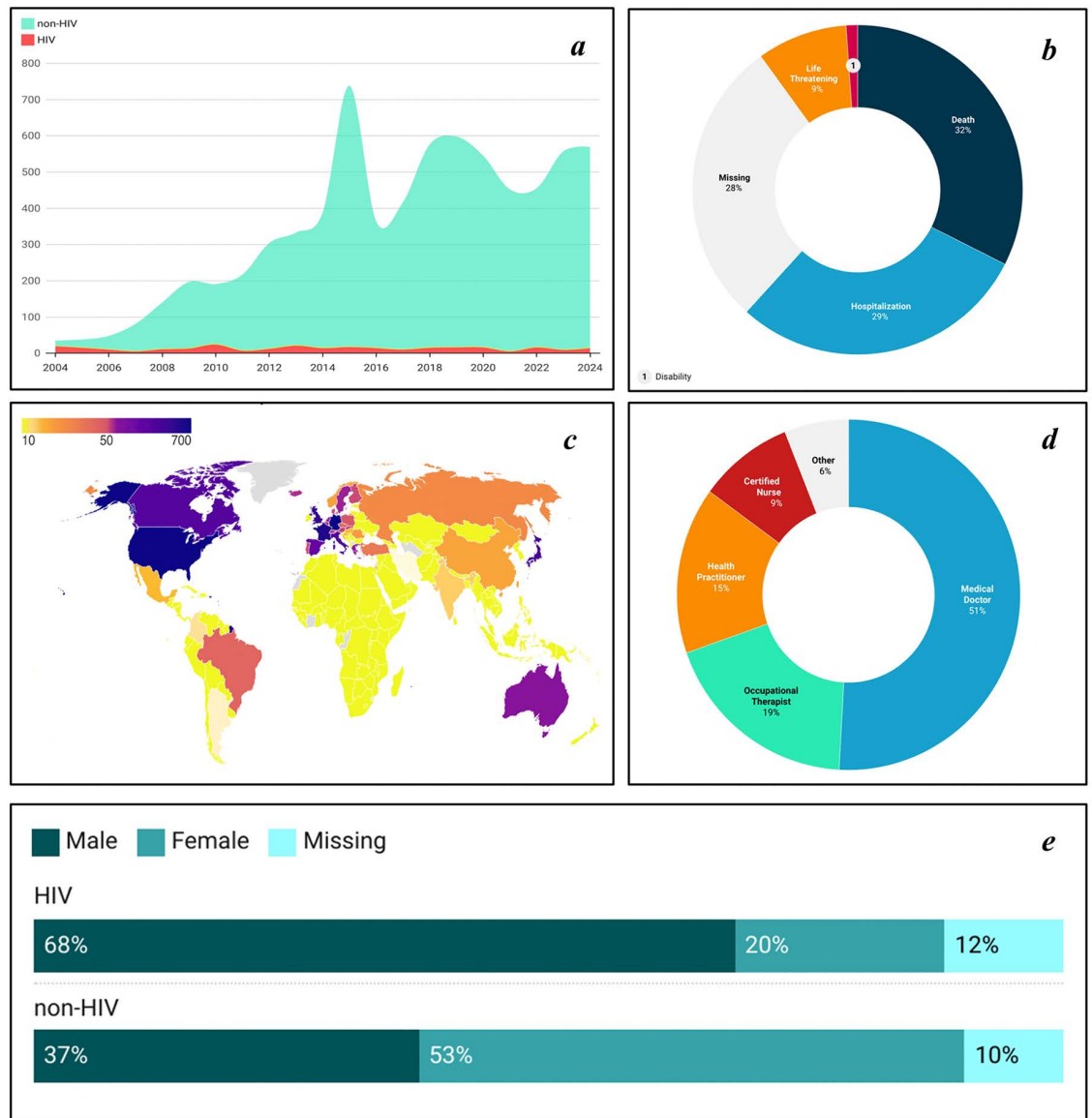

**Fig 2. Clinical and demographic characteristics of progressive multifocal leukoencephalopathy reports in FAERS (2004–2024).** a: Annual number of PML reports in HIV versus non-HIV populations. b: Clinical outcomes of PML reports, including proportions of life, death, hospitalization, disability, and missing data. c: Geographic distribution of PML reports. The map was created by the authors using Datawrapper (https://app.datawrapper.de/) with a public-domain base map from Natural Earth. d: Reporter category (e.g., physician, other healthcare professional, consumer). e: Proportion of HIV and non-HIV PML reports by sex. Abbreviations: HIV, human immunodeficiency virus.

immunosuppressed populations; (2) Applying multiple signal-detection methods to FAERS to identify 72 drugs with strong PML signals, with immunomodulatory agents and cytotoxic drugs displaying the strongest signals, enhancing clinical relevance;(3) High PML reporting risks are mainly concentrated in multiple sclerosis, lymphoma, autoimmune diseases, and organ transplantation, reflecting disease-specific medication patterns and heightened clinical vigilance;(4) Certain drugs showed prolonged PML reporting signals, highlighting the need for long-term monitoring.(5)These findings provide valuable references for pharmacovigilance and clinical risk alerting.

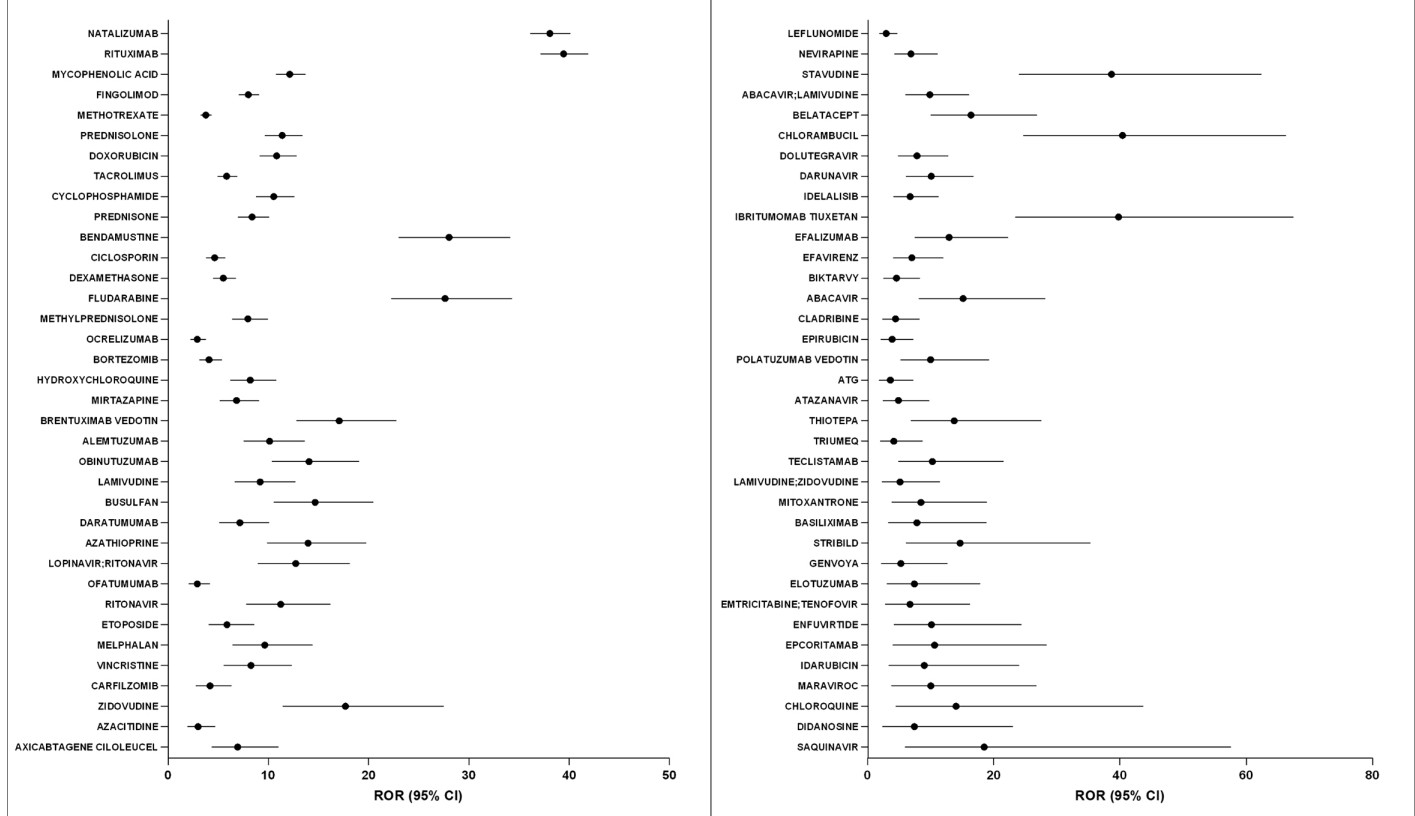

**Fig 3. Forest plot of RORs with 95% CI for 72 drugs positively associated with progressive multifocal leukoencephalopathy. BIKTARVY**, Bictegravir & Emtricitabine & Tenofovir Alafenamide; **ATG**, Antithymocyte Immunoglobulin; **STRIBILD**, Cobicistat & Elvitegravir & Emtricitabine & Tenofovir; **GENVOYA**, Cobicistat & Elvitegravir & Emtricitabine & Tenofovir Alafenamide. Abbreviations: ROR, reporting odds ratio; CI, confidence interval.

## 1. Trends in PML epidemiology and changes in population characteristics

PML was initially recognized as being most common among patients with intrinsic immune deficiencies, such as those with HIV/AIDS. With the widespread adoption of highly active antiretroviral therapy (HAART), the number of HIV-associated PML reports has stabilized at a relatively low level [6,12]. However, over the past two decades, the extensive use of a large number of immunosuppressive drugs—including biologics and immunomodulators—in the fields of cancer, autoimmune diseases, and organ transplantation has led to a rapid increase in drug-related PML [29–32]. The composition of PML reports in FAERS has shifted markedly, with HIV/AIDS-associated cases remaining low while drug-related PML reports have progressively increased to predominate (**Fig 2a**). This study reflects that drug-associated PML has become a new clinical focus for prevention and control. From the perspective of geographic distribution (**Fig 2c**), PML reports are predominantly concentrated in Europe and North America, likely reflecting stronger pharmacovigilance systems and widespread use of immunomodulatory agents in these regions [33], though reliance on MRI/cerebrospinal fluid JCV PCR for diagnosis [1] and the US-centric nature of the database may affect case confirmation and reporting completeness. Most reports in the FAERS database come from healthcare professionals (**Fig 2d**). This enhances the reliability of our data compared to spontaneous reporting systems that rely primarily on patient self-reporting. Regarding sex differences [34], we observed a higher proportion of female patients in the non-HIV population. This likely reflects underlying disease epidemiology and treatment patterns—particularly higher prevalence of systemic lupus erythematosus

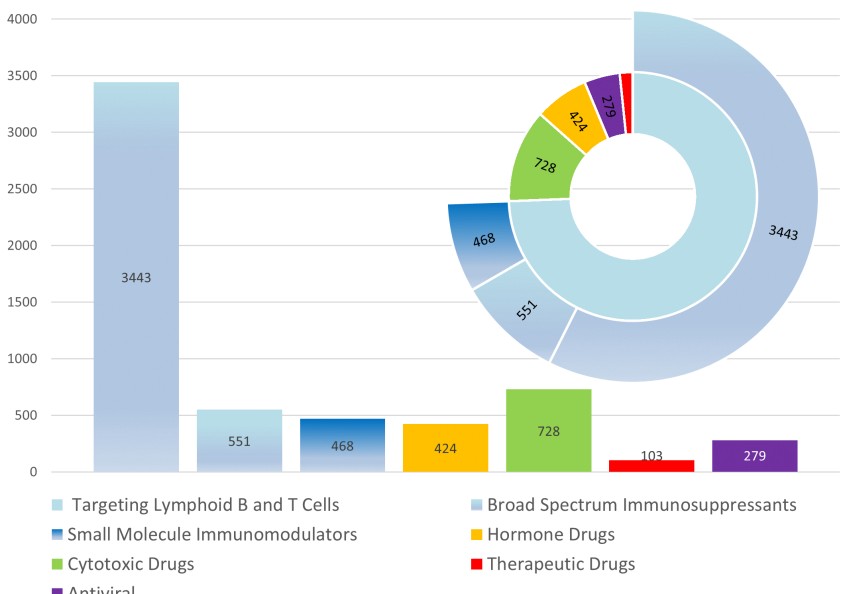

**Fig 4. Composition and distribution of 72 progressive multifocal leukoencephalopathy -associated drugs by pharmacological category.** The blue section indicates a major category, which includes three subcategories: drugs targeting lymphoid B and T cells, broad-spectrum immunosuppressants, and small molecule immunomodulators.

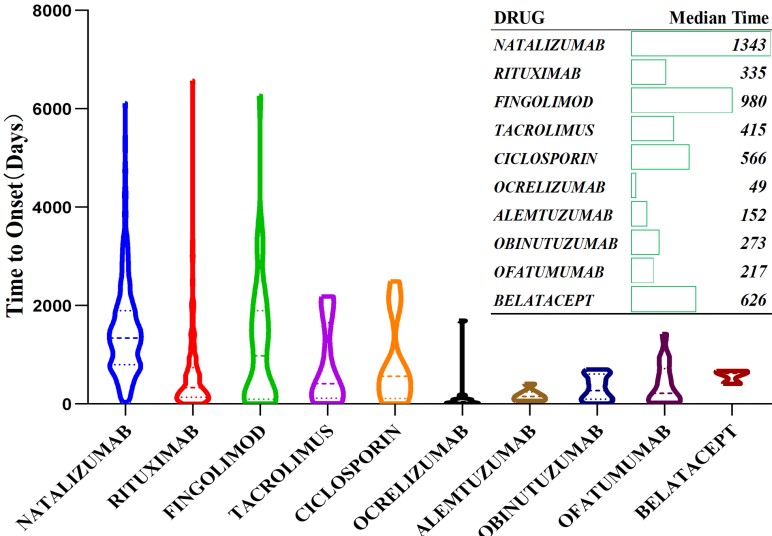

**Fig 5. Time-to-onset analysis of ten high-risk progressive multifocal leukoencephalopathy -associated drugs.**

and rheumatoid arthritis in women, leading to greater exposure to immunosuppressive/immunomodulatory therapies and more reported PML cases [35]—rather than a true sex-specific drug effect. Although sex-related biological factors in JCV infection and PML pathogenesis have been discussed [36], FAERS is subject to reporting bias, missing data, and

unmeasured confounding; our findings represent disproportionality signals and do not permit causal inference on sex-specific susceptibility.

## 2. Distribution of drug signals and analysis of high-risk indications

In this study, four mainstream signal detection methods (ROR, PRR, BCPNN, MGPS) were jointly used to determine risk drugs, including only those identified as positive by all four, thereby ensuring scientific rigor and reliability in the analysis of drug signals associated with PML. A total of 72 PML risk drugs were ultimately identified, mainly covering immunomodulators, cytotoxic drugs, biologic agents, and some novel targeted drugs (Fig 4). Our analysis revealed substantial heterogeneity in PML reporting odds ratios (RORs) among immunomodulatory agents, consistent with previous pharmacovigilance studies. Oshima et al. (2019) identified the highest reporting associations for natalizumab, while fingolimod showed markedly lower adjusted odds ratios [37]. Similarly, Honma et al. (2024) reported elevated RORs for B-cell–targeted agents such as rituximab, likely influenced in part by enhanced reporting awareness following established warnings [38]. Sharma et al. (2022) emphasized that carry-over effects from prior natalizumab exposure might contribute to PML occurrence during subsequent rituximab therapy [39]. Hence, FAERS-derived ROR signals should be interpreted as indicators of reporting association rather than absolute incidence, and integrated with clinical and epidemiological evidence for comprehensive risk evaluation.

Heatmap (S1 Fig) analysis of high-signal drugs and their indications showed that higher reporting rates of PML were highly concentrated in patients with multiple sclerosis, lymphoma, plasma cell myeloma, rheumatoid arthritis, and other immune-related diseases. S2 Fig further consolidates the top 30 diseases into 17 disease categories, also showing the correlation between immune and tumor-related diseases and PML. These patients are prone to immunosurveillance deficits either due to the diseases themselves or prolonged immunosuppressive therapy.

However, the observed reporting patterns are influenced by multiple biases: neurologists and hematologists have higher PML awareness and may report cases more frequently than rheumatologists or dermatologists; early drug safety warnings (e.g., natalizumab black box warning) may also have increased reporting [37,38]. Therefore, the concentration patterns reflect the combined result of true susceptibility, clinical practice, and reporting behavior, not equivalent to actual incidence risk distribution.

It is worth noting that among certain malignancies such as leukemia, PML may also occur even in the absence of immunosuppressant use because of persistent lymphocytopenia secondary to the disease itself [40,41]. This suggests that, in such cases, PML is often attributable to the synergistic immunosuppressive effects of both the underlying disease and drug therapy. By contrast, for patients with multiple sclerosis or post-organ transplant, PML more commonly results from long-term or intensive immunosuppressive treatment alone.

## 3. Pathogenic mechanisms and risk features of key drugs

While FAERS data are derived from a passive surveillance system and do not provide incidence estimates, we sought to explain high-signal drug–PML associations from a plausible pathogenic mechanisms perspective. Immunomodulators—such as rituximab and natalizumab—affect the body's immune status by acting on immune cells such as B/T cells. Previous studies have shown that low CD4+ T lymphocyte counts can be used for PML risk stratification [11,42]. However, natalizumab and rituximab, which ranked in the top 2 by ROR in our study, do not directly act on CD4+ T lymphocytes, and peripheral CD4+ T lymphocyte counts actually increase during natalizumab treatment [43], which seems inconsistent with previous research conclusions. Further review of the literature revealed that CD4+ T lymphocytes decrease during rituximab maintenance therapy, and CD4+ T counts show an upward trend when maintenance is discontinued, suggesting a synergistic effect between B/T cells [44–46]. Natalizumab-associated PML cases are time-dependent, and compared to CD4+ T cell counts, the decrease in CD4+ T lymphocyte bioenergetics (such as intracellular ATP) in natalizumab-associated PML cases may better reflect the body's immune status [47]. In addition, the decrease in the frequency of

specific CD4+ T subsets (central memory CD4+ T cells) is more closely associated with PML occurrence [48]. In summary, CD4+ T lymphocyte assessment should adopt a multidimensional strategy, fully considering CD4 counts, bioenergetic status (intracellular ATP concentration), and functional subsets.

Cytotoxic drugs—including alkylating agents, antimetabolites, cytotoxic chemotherapy drugs, etc.—induce cytotoxicity by interfering with DNA/RNA synthesis and replication, eliminating rapidly dividing cells, and significantly inhibiting the growth and proliferation of immune cells, causing sustained or profound immunosuppression [1,12]. Glucocorticoids suppress immune function via multiple mechanisms, with long-term or high-dose use markedly increasing PML risks [49].

HAART restores immunity and reduces opportunistic infections [50]. During this process, immune reconstitution may trigger immune reconstitution inflammatory syndrome (IRIS) [51–53], which is a paradoxical inflammatory response causing worsening of existing PML or new-onset disease with high morbidity and mortality [54].IRIS occurs during antiretroviral therapy or following immunosuppressant withdrawal [55].

In addition, this study has found associations between chloroquine, hydroxychloroquine, and mirtazapine with PML reports. Currently, there is no standard treatment for PML, although drugs such as mirtazapine, mefloquine, cidofovir, recombinant interleukin-7, interferon-α, and intravenous immunoglobulin (IVIG) are being explored [14,56,57]. Case reports and basic research suggest that JCV infection of oligodendrocytes depends on serotonin 2A receptor activity; thus, 5-HT2A receptor antagonists such as mirtazapine may be effective [58]. In vitro experimental evidence shows that mefloquine can inhibit JCV replication, though clinical efficacy remains limited [59,60]. Consequently, the principal goal in the treatment of PML remains restoring sufficient immune function to control JCV [1].

FAERS data cannot effectively disentangle drug effects from baseline risk (including underlying indication, disease severity, and concomitant immunosuppression), nor can it distinguish between therapeutic drugs and causative drugs; therefore, these signals should be viewed as hypothesis-generating rather than causal evidence, which constitutes an important limitation of this analysis.

## 4. Median time to onset (TTO) analysis of major drugs

We observed substantial heterogeneity in reported TTO (Fig 5), ranging from 49 days to 1343 days (approximately 1.6 to 44.8 months). Our findings differ from Maas's study reporting average TTO of 29.5 months (14.2–37.8 months in subgroups) [10].

Our relatively shorter TTO may be attributed to several factors. First, data incompleteness inherent in spontaneous reporting systems may affect the accuracy of temporal analyses, as highlighted by Hauben and Aronson [61], who demonstrated that missing data and reporting biases could significantly impact time-to-event estimates in pharmacovigilance databases.. Second, the heterogeneity in baseline diseases among our cohort likely influences PML onset timing, as Berger et al. [62]. Third, as illustrated in our analysis and consistent with Maas et al.'s findings [10], different immunosuppressive mechanisms produce varying time-to-onset patterns, with some drugs like natalizumab typically showing longer latency periods compared to other agents. Furthermore, Williamson and Berger [63] have described how different immunomodulatory mechanisms directly influence the timeline of PML development, with lymphocyte trafficking inhibitors showing distinct temporal patterns compared to B-cell depleting agents. Finally, differences in prior drug exposure may significantly impact TTO, as Schwab et al. [64] demonstrated that previous immunosuppressive treatment history can accelerate PML development, potentially explaining some of the shorter TTO reports in our cohort. In summary, TTO explanation needs to be more cautious.

## Conclusions and future perspectives

This study systematically analyzed the reporting patterns of drugs-associated PML based on the FAERS database, observing a significant shift in PML report composition over the past two decades: while HIV/AIDS-related case reports remained stable, reports of drug-induced immunosuppression-related reports continued to increase and became

predominant. This shift in reporting patterns coincides temporally with the widespread clinical use of biologics and immunomodulatory agents, suggesting that drug-induced PML has emerged as a critical focus for safety surveillance. The core value of this study lies in: systematically characterizing high-risk populations and disease distribution patterns (multiple sclerosis, lymphoma, autoimmune diseases, organ transplantation), providing guidance for clinical vigilance; revealing that different immunosuppressive mechanisms induce PML; identifying an extended risk time window (49–1343 days), challenging the adequacy of traditional short-term monitoring strategies; and suggesting that economically feasible monitoring indicators such as CD4+ cell counts and functional/subset features may be useful for risk stratification.

For patients receiving high-risk immunosuppressive therapy, clinical practice should integrate the following strategies: establish risk-stratified monitoring protocols based on JCV serology and CD4+ counts and functional/subset features; implement long-term surveillance extending beyond the initial treatment period; remain vigilant for IRIS-associated paradoxical worsening or new-onset PML during immune reconstitution (immunosuppressant discontinuation or HAART initiation); and balance immunosuppressive intensity with PML risk through individualized assessment.

Future research should focus on three main areas. First, multicenter prospective cohort studies should document detailed medication histories (monotherapy/combination regimens, cumulative doses, sequential order), establish clear exposed population denominators, quantify absolute incidence risks, and elucidate synergistic drug interactions; Second, advance mechanistic research and biomarker discovery to improve risk stratification and early screening protocols, for example, clarify immunologic thresholds for JCV reactivation and host genetic susceptibility, develop and validate risk prediction models integrating multidimensional indicators, which remains crucial for translating pharmacovigilance findings into clinical practice. Third, develop long-term monitoring approaches that address the time-dependent nature of PML risk through Integrating real-world data, international data sharing and exploration of JCV-targeted prevention strategies to establish evidence-based long-term monitoring guidelines.

## Limitations

This study has inherent limitations. (1) The geographic concentration of PML reports in European and American regions may reflect variations in FAERS utilization and medical surveillance infrastructure, potentially limiting global representativeness. (2) As a spontaneous reporting system, FAERS has inherent flaws, including underreporting, incomplete information, and selection bias, which may lead to underestimation or overestimation of certain signals [65]. (3) The absence of key clinical variables (concomitant medications, detailed underlying conditions, and treatment dosage and duration) represents a significant limitation [66], with TTO potentially underestimating true cumulative exposure. (4) In addition, the detected signals inevitably mix potential drug–PML associations with the background PML risk conferred by the underlying indications (particularly HIV/AIDS and lymphoma), making confounding by indication difficult to disentangle in this dataset Although we employed multiple signal detection methods to enhance reliability, differences in sensitivity and specificity across methods may lead to inconsistencies in results. Therefore, our findings should be considered alongside evidence from clinical trials, cohort studies, and other real-world data sources to form a more comprehensive understanding.

## Supporting information

**S1 Fig. Heatmap of the top 30 drugs and 30 diseases associated with PML.**
(TIF)

**S2 Fig. Ranking of disease categories associated with PML cases after classification merger.**
(TIF)

**S1 Table. List of 72 drugs with positive PML signals in all four detection algorithms, ranked by case numbers.**
(DOCX)

## Author contributions

**Conceptualization:** Hailong Wang, Lijuan Shangguan.

**Formal analysis:** Xinyi Li.

**Methodology:** Hailong Wang.

**Project administration:** Lijuan Shangguan.

**Software:** Hailong Wang.

**Supervision:** Xinyi Li, Lijuan Shangguan.

**Validation:** Xinyi Li.

**Writing – original draft:** Lijuan Shangguan.

**Writing – review & editing:** Hailong Wang, Lijuan Shangguan.

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
