## [Decision Letter · Decision Letter 0]

23 Oct 2025

Disproportionality Analysis of Drug-Associated Progressive Multifocal Leukoencephalopathy Using Spontaneous Reports: A 20-Year Signal Detection Study Based on the FAERS Database

PLOS ONE

Dear Dr. shangguan,

Thank you for submitting your manuscript to PLOS ONE. After careful consideration, we feel that it has merit but does not fully meet PLOS ONE’s publication criteria as it currently stands. Therefore, we invite you to submit a revised version of the manuscript that addresses the points raised during the review process.

We look forward to receiving your revised manuscript.

Kind regards,

Mehmet Baysal

Academic Editor

PLOS ONE

Journal Requirements:

3. Thank you for uploading your study's underlying data set. Unfortunately, the repository you have noted in your Data Availability statement does not qualify as an acceptable data repository according to PLOS's standards.

Additional Editor Comments:

The reviewers noted strengths of your work  but also several critical deficiencies. The main parts of the manuscript that need to be strengthened are discussion, limitations and conclusion. Pay very close attention to the recommendations of the reviewers as you revise.

Reviewer's Responses to Questions

**Comments to the Author**

1. Is the manuscript technically sound, and do the data support the conclusions?

Reviewer #1: Partly

Reviewer #2: Partly

2. Has the statistical analysis been performed appropriately and rigorously?

Reviewer #1: Yes

Reviewer #2: Yes

3. Have the authors made all data underlying the findings in their manuscript fully available?

Reviewer #1: Yes

Reviewer #2: Yes

4. Is the manuscript presented in an intelligible fashion and written in standard English?

Reviewer #1: Yes

Reviewer #2: Yes

Reviewer #1: The manuscript presents an analysis of very interesting data that is worth adding to the published literature. The authors fulfill many recommendations for this article type (Fusaroli et al, Reporting of a Disproportionality Analysis for Drug Safety Signal Detection using Individual Case Safety Reports in PharmacoVigilance (READUS-PV): Development and Statement. Drug Saf 2024 May 7;47(6):575-584). Nevertheless, the manuscript itself has issues that warrant substantial rewriting and then re-review.

The authors analyze 20 years of data from the US FDA Adverse Events Reporting System (FAERS) to demonstrate the association of PML with medications. The description of the methods was reasonable; however there was not detailed information on how reports were excluded other than duplication. In figure 1, the arrows from the DEMO (after de-duplication), DRUG and REAC modules should all point together to the final dataset. The core analyses use 4 methods to quantify the level of association of medications with PML in an effort to avoid the bias of any one method (Reporting Odds Ratio, Proportional Reporting Ratio, Bayesian confidence propagation neural network [BCPNN], and multi-item

74 gamma Poisson shrinker [MGPS]). All of these methods are on use by the US FDA and WHO and by researchers to analyze large data sets for drug and vaccine adverse events. They each have different characteristics. The authors chose to require that all 4 measures indicate an association as a conservative method to identify drug-PML associations. I cannot comment on the individual measures but was impressed with their requirement of agreement among the four. The authors report 72 drugs have significant associations with PML with the criterion that all 4 measures fulfill pre-defined levels. The four drugs with the strongest associations are descried in the text; details on the 4 measures for all 72 drugs are listed in Supplemental Table 1. It would be better to include Table 1, or at least part of the table with the highest impacted drugs, as part of the publication. Limited information regarding the demographics are provided (figures 2d and 2e). The graph of the time course of reports is interesting (figure 2a). The forest plot of RORs for the 72 medications (figure 3) is very interesting. The text mentions the strength of the ROR association for only a few medications; however no comment is made in the text about the widely varying strength of associations across the 72 medications nor is there an interpretation of the great differences in ROR confidence intervals. Figure 5, graphically displaying the time course from drug initiation to PML diagnosis from the limited information in the FAERs case reports, is interesting but likely to be misleading because information on prior use and duration of other potentially relevant medications is not described and may not be available.

The two main concerns with this manuscript are (1) the inherent bias of the data source and (2) the lack of a conclusion drawn from their analysis.

Regarding the data source, the authors make a partial acknowledgement of bias in the section on limitations. However, the text suggests that the incidence of events over 20 years is evident from FAERS, whereas this is just the incidence of reports. They cannot comment on population incidence of HIV associated or non-HIV associated PML from their data. The authors acknowledge the FAERS data are mostly from North America and Europe and comment that is likely due to geographic limitations on medication access; while that is true there are also likely geographic limitations on ability to diagnose PML (MRI scans, CSF JC virus nucleic acid amplification testing) and use of the US English language database.

Another potential bias in the database is that PML may be more likely to be diagnosed by neurologists treating multiple sclerosis and hematologists treating lymphoma than rheumatologists or dermatologists using immunosuppressive therapies. Another potential bias in a database of spontaneous case reports is that there was a very early warning about the association of certain drugs. For example, early reports linked natalizumab with PML leading to the drug being unlicensed temporarily by the FDA then reinstated with a black box warning. Once this association was suspected, voluntary reporting of PML diagnosis among persons treated with this drug may have become more frequent.

In the conclusion section, the authors write extensively about HIV associated with PML and IRIS. This is misplaced. HIV induced immunosuppression was a cause of PML before there were any antiretroviral medications. The discussion of HIV associated PML is not relevant except for the observation that reports of PML in FAERS were few and stable during the 20 years when reports of drug associated PML increased dramatically.

There is not much to say about steroids and risk of PML. A very few cases were reportedly associated with steroids which is remarkable when one considers how widespread is the use of therapeutic systemic corticosteroids.

Importantly, I do not find a new conclusion based on the data. Reports of drug associated PML have increased over 20 years, some of the PML patients had immunosuppressive disorders and almost all the rest received medications that were immunosuppressive. I did not find where the authors offered a new hypothesis derived from their analysis; that would be appropriate to justify publishing this manuscript.

Reviewer #2: The paper presents data on an adverse drug database, linking the PML infection to the use of immunosuppressant drugs and describing demographic and drug data of these adverse drug reactions. Although it is a well-conducted study with a large dataset, the manuscript lacks a robust discussion that can demonstrate the importance and relevance of its findings. Furthermore, the figures presented in the manuscript are of poor quality, making it difficult to read their captions and fully understand them.

The methods used to analyze the adverse drug reactions should be fully described in the manuscript, and their use discussed as a way to provide clinical and policy evidence. I believe the same applies to the validation methods cited in lines 93-94 for the case identification. It is important to ensure reproducibility of the research.

The discussion section focuses a lot on the drugs and their mechanisms of action. Instead, it should focus on the findings' implications for practice, public health policy and research. It would also be important to relate and contrast it with other research in the field that either uses similar methods, drugs, or problems to contextualize your research in the scientific field. How does it advance the research in the theme? How is it similar or different from the ones already published? How can these differences be explained? What are the manuscript's main contributions?

As minor corrections, I suggest reviewing the points listed above:

• Line 59: Correct reference format

• Lines 60-63: A reference should be included to support it.

• Lines 86: The link should be included as a reference.

• The quality of the figures is not very good. It is not possible to read the captions in the figures, for instance, in 2a, to identify which color represents HIV or non-HIV PML infection. Neither is it possible to read the years in x. The same applies to Figures 2b and 2d; it is not possible to read what is in the graphs. In Figure 3, it is not possible to read the names of the drugs, even when using zoom.

• I do not understand why there are two types of graphs for the same numbers in Figure 4.

• Lines 227-228: Your data do not support this statement: “and that the use of immunomodulatory drugs is more prevalent”. This could be a hypothesis and along with a reference on drug use, that may indicate it

• Lines 231-233: It is important to include a reference that expresses that autoimmune diseases are more prevalent in women. The same applies to the sentence in lines 234-236.

• Lines 377-378: “for drugs with unclear mechanisms that showed positive signals in our analysis but lack established causal relationships”: which drugs fall into this? Shouldn´t it be stated and discussed along with the results?

**Do you want your identity to be public for this peer review?** For information about this choice, including consent withdrawal, please see ourPrivacy Policy

Reviewer #1: No

Reviewer #2: No

---

## [Author Response · Author response to Decision Letter 1]

1 Dec 2025

Responses to Academic Editor’s Comments

Additional Editor Comments:

The reviewers noted strengths of your work but also several critical deficiencies. The main parts of the manuscript that need to be strengthened are discussion, limitations and conclusion. Pay very close attention to the recommendations of the reviewers as you revise.

Comment 1:

“The main parts of the manuscript that need to be strengthened are discussion, limitations and conclusion.”

Response:

We sincerely appreciate the editor’s valuable summary and guidance. In this revision, we have substantially strengthened the Discussion, Limitations, and Conclusion sections:

The Discussion section has been reorganized to reduce excessive descriptions of drug mechanisms and to enhance comparison of ROR findings with previous literature, with an emphasis on the implications of the results for pharmacovigilance practice, regulatory decision making, and future research directions.

The Conclusion section has been rewritten to condense the main findings, clarify that FAERS data reflect reporting trends rather than actual incidence, and propose new hypotheses regarding the monitoring of drug associated PML.

The Limitations section now more explicitly addresses the inherent potential biases in the FAERS database, including reporting bias, diagnostic and regional limitations, and differences among clinical specialties in identifying PML cases.

Comment 2:

“Please ensure that your manuscript meets PLOS ONE’s formatting requirements.”

Response:

We have reformatted the entire manuscript by following the official PLOS ONE style templates and have carefully checked file naming conventions for all submissions, including the main manuscript, marked up revision file, and supplementary materials.

Comment 3:

“PLOS requires an ORCID iD for the corresponding author.”

Response:

The corresponding author’s ORCID iD has been verified and linked through the Editorial Manager system as required.

Comment 4:

“The data repository noted in your Data Availability Statement does not qualify as an acceptable repository according to PLOS’s standards.”

Response:

In response, we have now deposited the minimal dataset necessary for replication of our study findings in the public repository Figshare (DOI: https://doi.org/10.6084/m9.figshare.30575225) and have rewritten Data Availability Statement.

The uploaded dataset includes:

The list of 7,244 report IDs related to progressive multifocal leukoencephalopathy (PML) (as identified from the deduplicated DEMO table);

The list of 229 unique drugs included in our disproportionality analysis;

Results from the four signal detection methods applied to these drugs.

We have also updated the Data Availability Statement in the revised manuscript accordingly.

Comment 5:

“Please pay close attention to the reviewers’ recommendations.”

Response:

We have carefully addressed every point raised by both reviewers. Major revisions include reorganizing the discussion to strengthen scientific interpretation, refining data presentation and figure quality, clarifying methodological details, and expanding the explanation of data limitations. Comprehensive point by point responses are provided below.

Responses to Reviewer’s Comments

Reviewer #1:

Comment 1:

The manuscript presents an analysis of very interesting data that is worth adding to the published literature. The authors fulfill many recommendations for this article type (Fusaroli et al, Reporting of a Disproportionality Analysis for Drug Safety Signal Detection using Individual Case Safety Reports in PharmacoVigilance (READUS-PV): Development and Statement. Drug Saf 2024 May 7;47(6):575-584). Nevertheless, the manuscript itself has issues that warrant substantial rewriting and then re-review.

Response:

We sincerely thank the reviewer for recognizing the value of our study. In response to your suggestions, we have thoroughly revised the manuscript to make it clearer and easier to understand. We have objectively presented our findings, clarified the conclusions, appropriately applied FAERS database data, distinguished between incidence and reporting rates, and removed any potentially confusing or misleading statements.

Comment 2:

The authors analyze 20 years of data from the US FDA Adverse Events Reporting System (FAERS) to demonstrate the association of PML with medications. The description of the methods was reasonable; however there was not detailed information on how reports were excluded other than duplication. In figure 1, the arrows from the DEMO (after de-duplication), DRUG and REAC modules should all point together to the final dataset.

Response:

Thank you very much for this helpful comment. In response, we have completely redrawn Figure 1 to better illustrate the relationship among the DEMO, DRUG, and REAC modules in the data integration process. The revised flowchart now clearly shows that all three modules were merged after the de duplication step to generate the final dataset for screening PML related reports.

Regarding the exclusion criteria, we have clarified in the Methods section that, apart from removing duplicate cases, we directly included all records containing the preferred term (PT) “progressive multifocal leukoencephalopathy (PML)” as identified in the REAC module. This revision improves the transparency of our data selection process and aligns the text with the updated Figure 1.

Comment 3:

The core analyses use 4 methods to quantify the level of association of medications with PML in an effort to avoid the bias of any one method (Reporting Odds Ratio, Proportional Reporting Ratio, Bayesian confidence propagation neural network [BCPNN], and multi-item gamma Poisson shrinker [MGPS]). All of these methods are on use by the US FDA and WHO and by researchers to analyze large data sets for drug and vaccine adverse events. They each have different characteristics. The authors chose to require that all 4 measures indicate an association as a conservative method to identify drug-PML associations. I cannot comment on the individual measures but was impressed with their requirement of agreement among the four. The authors report 72 drugs have significant associations with PML with the criterion that all 4 measures fulfill pre-defined levels. The four drugs with the strongest associations are descried in the text; details on the 4 measures for all 72 drugs are listed in Supplemental Table 1. It would be better to include Table 1, or at least part of the table with the highest impacted drugs, as part of the publication. Limited information regarding the demographics are provided (figures 2d and 2e).

Response:

We have incorporated the list of the 72 drugs with significant associations directly into the main text (previously contained in the Supplement). Given the space limitations, we highlighted the subset of drugs with the strongest signals in Table 1. Regarding demographic data, we have clarified in the manuscript that the FAERS dataset provides limited demographic dimensions, which inherently restricts the depth of such analyses.

Comment 4:

The graph of the time course of reports is interesting (figure 2a). The forest plot of RORs for the 72 medications (figure 3) is very interesting. The text mentions the strength of the ROR association for only a few medications; however no comment is made in the text about the widely varying strength of associations across the 72 medications nor is there an interpretation of the great differences in ROR confidence intervals.

Response:

We appreciate this insightful suggestion. We have revised the Discussion section to provide a detailed interpretation of the varying strengths of association and the wide confidence interval ranges observed for the ROR values across different drugs. We now discuss how these differences may reflect variable reporting frequencies, therapeutic contexts, and prior awareness of PML risk.

Comment 5:

Figure 5, graphically displaying the time course from drug initiation to PML diagnosis from the limited information in the FAERs case reports, is interesting but likely to be misleading because information on prior use and duration of other potentially relevant medications is not described and may not be available.

Response:

Thank you for this important observation. We acknowledge that while we found this aspect of the analysis interesting, we indeed overlooked its potential to mislead readers. We have made substantial revisions to address this concern. Specifically, we have added a comprehensive discussion of the limitations and inherent biases of the TTO analysis, ensuring that readers are fully informed of the constraints in interpreting these findings.

Comment 6:

The two main concerns with this manuscript are (1) the inherent bias of the data source and (2) the lack of a conclusion drawn from their analysis.

Regarding the data source, the authors make a partial acknowledgement of bias in the section on limitations. However, the text suggests that the incidence of events over 20 years is evident from FAERS, whereas this is just the incidence of reports. They cannot comment on population incidence of HIV associated or non-HIV associated PML from their data. The authors acknowledge the FAERS data are mostly from North America and Europe and comment that is likely due to geographic limitations on medication access; while that is true there are also likely geographic limitations on ability to diagnose PML (MRI scans, CSF JC virus nucleic acid amplification testing) and use of the US English language database.

Response:

We appreciate the reviewer’s comment and fully concur. We have substantially revised the Limitations section to clarify that our findings reflect FAERS reporting frequencies rather than true disease incidence. We explicitly acknowledge that FAERS data are subject to underreporting bias, differential reporting across specialties, and regional variation, and therefore cannot be used to estimate population-level incidence. In addition, we have replaced all instances of the term “cases” with “reports” throughout the manuscript to avoid any potential misunderstanding by readers. Following your guidance, these revisions significantly strengthen the scientific rigor and transparency of our manuscript.

Comment 7:

Another potential bias in the database is that PML may be more likely to be diagnosed by neurologists treating multiple sclerosis and hematologists treating lymphoma than rheumatologists or dermatologists using immunosuppressive therapies. Another potential bias in a database of spontaneous case reports is that there was a very early warning about the association of certain drugs. For example, early reports linked natalizumab with PML leading to the drug being unlicensed temporarily by the FDA then reinstated with a black box warning. Once this association was suspected, voluntary reporting of PML diagnosis among persons treated with this drug may have become more frequent.

Response:

This is an excellent and realistic point. We have expanded the Discussion section to discuss how professional specialization, disease focus, and prior regulatory alerts (e.g., the early natalizumab signal) may have influenced reporting frequency. These contextual factors have been acknowledged as potential sources of reporting bias in our interpretation.

Comment 8:

In the conclusion section, the authors write extensively about HIV associated with PML and IRIS. This is misplaced. HIV induced immunosuppression was a cause of PML before there were any antiretroviral medications. The discussion of HIV associated PML is not relevant except for the observation that reports of PML in FAERS were few and stable during the 20 years when reports of drug associated PML increased dramatically.

Response:

Thank you for this important clarification. We have removed the lengthy section on HIV and IRIS from the Discussion. A brief description of the clinical context of IRIS is now provided. In parallel, by referencing the 20-year trend in PML reports extracted from the FAERS database, we have made it clear that the number of reports of HIV-associated progressive multifocal leukoencephalopathy (PML) has remained relatively stable over time, in stark contrast to the marked increase in drug-associated PML reports. This sharp contrast is now explicitly highlighted in the revised manuscript.

Comment 10:

There is not much to say about steroids and risk of PML. A very few cases were reportedly associated with steroids which is remarkable when one considers how widespread is the use of therapeutic systemic corticosteroids. Importantly, I do not find a new conclusion based on the data. Reports of drug associated PML have increased over 20 years, some of the PML patients had immunosuppressive disorders and almost all the rest received medications that were immunosuppressive. I did not find where the authors offered a new hypothesis derived from their analysis; that would be appropriate to justify publishing this manuscript.

Response:

we have revised the Conclusion to emphasize that drug induced PML has become a key focus of pharmacovigilance. This study systematically characterizes high risk populations and disease patterns (including multiple sclerosis, lymphoma, autoimmune diseases, and organ transplantation), reveals that distinct immunosuppressive mechanisms—lymphocyte depletion and interference with DNA synthesis—converge on impaired immune surveillance pathways leading to PML, and identifies an extended latency window (49–1343 days) that challenges traditional short term monitoring strategies. We further propose that practical biomarkers such as CD4⁺ cell counts may facilitate risk stratification in clinical settings.

Reviewer #2:

Comment 1:

The paper presents data on an adverse drug database, linking the PML infection to the use of immunosuppressant drugs and describing demographic and drug data of these adverse drug reactions. Although it is a well-conducted study with a large dataset, the manuscript lacks a robust discussion that can demonstrate the importance and relevance of its findings.

Response:

We have thoroughly rewritten the Discussion to better highlight the importance and relevance of our findings. The revised text now focuses on the implications for pharmacovigilance practice, clinical decision making, and public health policy, and includes clear comparisons with previous studies addressing similar topics.

Comment 2:

Furthermore, the figures presented in the manuscript are of poor quality, making it difficult to read their captions and fully understand them.

Response:

All figures (Figures 1–5) have been fully revised and re exported at high resolution (300 dpi). Following PLOS ONE’s official requirements, we used the Art Analysis (PageMajik) online tool provided by the journal to verify image resolution, font embedding, and color compliance. All figures passed the quality check and now comply with PLOS ONE’s technical specifications. Please note that the figures may appear slightly compressed in the generated PDF due to the system’s automatic optimization process; we kindly ask you to refer to the original image files for full resolution quality.

Comment 3:

The methods used to analyze the adverse drug reactions should be fully described in the manuscript, and their use discussed as a way to provide clinical and policy evidence. I believe the same applies to the validation methods cited in lines 93-94 for the case identification. It is important to ensure reproducibility of the research.

Response:

We appreciate the reviewer’s valuable comment regarding the need for clear methodological description and reproducibility. In the revised manuscript, we have clarified the references to the established methodologies cited in lines 93–94, which are widely recognized in pharmacovigilance research and validated in prior publications. These references provide the detailed algorithms and validation procedures for adverse event case identification and signal detection.

To further improve transparency, we have refined the workflow figure to illustrate the screening and exclusion process more specifically and deposited the minimal dataset required to replicate our findings in the public repository Figshare (DOI: 10

---

## [Decision Letter · Decision Letter 1]

22 Dec 2025

Dear Dr. shangguan,

Thank you for submitting your manuscript to PLOS ONE. After careful consideration, we feel that it has merit but does not fully meet PLOS ONE’s publication criteria as it currently stands. Therefore, we invite you to submit a revised version of the manuscript that addresses the points raised during the review process.

We look forward to receiving your revised manuscript.

Kind regards,

Mehmet Baysal

Academic Editor

PLOS One

Journal Requirements:

Additional Editor Comments:

The reviewers are generally pleased with the corrections and responses you've provided, but there are still some suggestions/quotes that need correction and clarification.

Reviewers' comments:

Reviewer's Responses to Questions

**Comments to the Author**

Reviewer #1: (No Response)

Reviewer #2: All comments have been addressed

2. Is the manuscript technically sound, and do the data support the conclusions?

Reviewer #1: Partly

Reviewer #2: Yes

3. Has the statistical analysis been performed appropriately and rigorously?

Reviewer #1: Yes

Reviewer #2: Yes

4. Have the authors made all data underlying the findings in their manuscript fully available?

Reviewer #1: Yes

Reviewer #2: Yes

5. Is the manuscript presented in an intelligible fashion and written in standard English?

Reviewer #1: Yes

Reviewer #2: Yes

Reviewer #1: The authors have done substantial analysis of an enormous dataset and provide interesting results of their analysis. However, in this second submission they still have not clearly and succinctly stated the specific contribution of their analysis to medical knowledge. For publication in a general medical journal this is essential for the readers; it may be less necessary for publication in a journal dedicated to pharmacovigilance.

Strengths: The results in Table 1, Figure2a, figure 3, and S1 figure (heat map) and Fig 5 are very interesting and worth publication.

Problems:

There are many studies of medications that are associated with PML and are likely causes of PML. In my view, the authors do not explain clearly what their study adds to current knowledge. Disproportionality analysis of large databases such as FAERS is subject to many types of bias so is most useful for generation of hypotheses. The associated case reports often lack detail so that causality rarely can be convincingly demonstrated with this type of data. The authors acknowledge the many types of bias and the lack of proof of causality in their study.

Based on my brief literature review, this study is unusual in analyzing data from the FAERs database over a long (20 year) and recent (up to 2024) time frame, and in analyzing all drugs with a reported PML association regardless of the drug type or underlying health condition. Other disproportionality studies of PML within FAERS were either done longer ago, used a shorter time frame, or were restricted to specific underlying conditions. This may be the largest published recent disproportionality analysis of PML as an adverse drug event, although there may be other similar analyses undergoing review (not by me).

The authors should tell the reader what this study adds to current knowledge. Perhaps their list of drug associations is longer than other published studies but the authors do not make this claim. Perhaps the authors found novel associations, but they do not make this claim. Perhaps the authors, requiring concurrence among 4 disproportionality methods, did not find convincing evidence of a PML association for some drugs previously linked with PML, however the authors did not make this claim.

The authors do not address the issue of separating the association of PML with a drug from the possible association of a drug with the underlying condition for which the drug was prescribed. HIV associated PML presents a particular problem. The comparison of reports of PML associated with HIV versus non-HIV conditions is useful. The rest of their discussion regarding HIV is not clear. Several antiretroviral medications are listed in this analysis. Only maraviroc has a potential for direct immune modulation so the association of other antiretrovirals (lamivudine, lopinavir, ritonavir, etc.) is through the underlying condition. Most recent HIV-associated PML is in persons new to treatment, undertreated, or is a result of immune reconstitution inflammatory syndrome (IRIS) PML due to effective antiretroviral treatment. This is very different conceptually from the role of natalizumab or rituximab in PML. In this analysis, in which all drug associations are analyzed, it is important for the authors to distinguish these types of medication. Less extreme is the quantifying the association of medications in lymphoma treatment given the underlying risk of lymphoma itself with PML. If the FAERS data cannot address this, the authors should mention this as a limitation.

The study uses 4 disproportionality methods for identifying reports of PML as an AE drug reports but does not explain how these are different, what results of each mean, and why the combination they use is better than other studies. Although these are methods used for pharmacovigilance studies, it would strengthen this manuscript submitted to a general medical journal to include a brief rationale and a brief explanation of the strengths of the different methods.

The sex difference on drug-PML associations between HIV and non-HIV is almost certainly due to the epidemiology of the underlying conditions and no inference about sex differences in drug-associated PML is justified. In the US and Europe most HIV disease occurs among men. Most autoimmune disease occurs among women.

The authors state CD4 measurements can help determine risk of drug associated PML but they present no evidence for this and other information makes this assertion unlikely. Rituximab, the drug with the second most frequent associations and the second highest ROR, impacts B lymphocytes cells and not CD4 T lymphocytes. Natalizumab, the drug with the most frequently reported PML associations and the highest ROR, as is thought to work on multiple sclerosis by limiting lymphocyte migration across the blood brain barrier. There are published data demonstrating that CD4 lymphocyte counts increase during natiluzumab treatment (Berkovich R, Togasaki D, Yen S, Steinman L. Ann Clin Transl Neurol. . 2015 May;2(5):570-4. doi: 10.1002/acn3.190. Epub 2015 Mar 6.)

Reviewer #2: The careful revision is appreciated, and the Discussion section has shown significant improvement. There are now only a few minor corrections remaining:

1) Figure 2 Legends: The legends for Figure 2 are incorrect. Specifically, the legend for Figure 2c is mistakenly applied to Figure 2d, as 2c appears to concern geographic distribution reporting, while 2d relates to professional category reporting.

2) Missing Citations: The second paragraph of the Discussion, Section 3 (lines 323-330), requires a reference citation.

3) Punctuation Review: The third paragraph of the Conclusion section and the Limitations section need to be reviewed for punctuation errors.

**Do you want your identity to be public for this peer review?** For information about this choice, including consent withdrawal, please see our Privacy Policy

Reviewer #1: No

Reviewer #2: **Yes:** Luisa Arueira Chaves

---

## [Author Response · Author response to Decision Letter 2]

31 Dec 2025

Reviewer #1

Comment 1) The manuscript does not clearly and succinctly state the specific contribution to medical knowledge.

Response: Thank you for this helpful suggestion. We have revised both the Introduction and the opening paragraph of the Discussion to consistently and explicitly state the study’s key contributions. Specifically, we clarify that our analysis uses FAERS data from 2004–2024 (updated to 2024Q4), provides an all-drug, indication-agnostic panoramic screen of drug–PML reporting, and applies four complementary disproportionality algorithms (ROR, PRR, BCPNN, and MGPS) with a concordance-only signal definition to improve specificity, yielding an updated and conservative signal inventory for prioritizing further validation. (Introduction, last paragraph).________________________________________

Comment 2) Confounding by indication: difficulty separating drug–PML association from underlying conditions (esp. HIV, lymphoma).

Response: Thank you for pointing out this key interpretive issue. A major feature of our study is the systematic identification of signals for all drugs reported with PML over an extended FAERS time window. We acknowledge that these signals inevitably conflate potential drug–PML associations with the background PML risk attributable to the underlying indications (particularly HIV and lymphoma), as well as drugs implicated in PML pathogenesis and drugs used to treat PML. Given the limited clinical detail and incompleteness of FAERS reports, we cannot reliably disentangle these effects at the individual level. We have therefore added an explicit statement in the last paragraph of the Discussion to highlight this limitation and to frame our findings as hypothesis-generating rather than causal (Discussion, 3. Pathogenic Mechanisms and Risk Features of Key Drugs).________________________________________Comment 3) Explain the four disproportionality methods and why requiring concordance is beneficial.

Response: We have revised the beginning of the “Signal detection methods” description to briefly clarify the conceptual differences among the four algorithms and the rationale for combining them. Specifically, we added 2–3 sentences to explain that ROR/PRR are frequentist disproportionality measures based on 2*2 contingency tables and may be sensitive to sparse counts, whereas BCPNN/MGPS are Bayesian shrinkage approaches that stabilize estimates for rare drug–event pairs. We also clarified that we required concordant signals across all four methods to prioritize specificity and reduce spurious findings, while acknowledging a potential loss of sensitivity. (Materials and methods, 1. Signal Detection Methods)

Comment 4) Sex differences: avoid inferring sex-specific susceptibility; differences likely reflect epidemiology of underlying conditions.

Response: We agree. We revised the Discussion to avoid inferring sex-specific susceptibility. We now interpret the observed sex distribution as more likely reflecting the epidemiology of underlying diseases and treatment patterns (e.g., HIV predominance among men; autoimmune diseases predominance among women), and we explicitly note that FAERS disproportionality signals do not permit causal inference regarding sex-specific risk. (Discussion, 1.Trends in PML Epidemiology and Changes in Population Characteristics, last paragraph)________________________________________Comment 5) CD4 statements are too strong / unsupported; not applicable to rituximab/natalizumab contexts.

Response: Thank you for raising this important point. We fully agree that rituximab does not directly target CD4+ T lymphocytes and that peripheral CD4+ T lymphocyte counts may increase during natalizumab therapy.We would like to clarify that our statement regarding "CD4+ T lymphocyte measurements may help assess the risk of drug-associated PML" was not intended to present CD4+ T lymphocytes as a direct "risk biomarker" or as causal evidence. Rather, our rationale is based on the core pathogenesis of PML—reactivation of JCV in the context of immunosuppression or immune reconstitution. Within this framework, the number and functional status of CD4+ T lymphocytes can, to some extent, indirectly reflect overall immune competence and therefore serve as a practical "barometer" of immune status. Even when certain drugs do not act directly on CD4 (e.g., rituximab primarily depletes B cells), CD4-related immune dysfunction may still occur due to the underlying disease, immune cell interactions, or sustained immunosuppression. Therefore, from this perspective, CD4+ T lymphocyte-centered assessments—including counts, functional status evaluation, and specific subset characteristics—may serve as part of supportive risk stratification, rather than being used in isolation to draw conclusions. Your suggestions have helped us better refine our statement.

Regarding the apparent paradox of "increased CD4+ T lymphocyte counts" with natalizumab, we carefully reviewed the corresponding article you provided and incorporated its findings into our manuscript. However, we noted that the article suggests the increase in peripheral blood CD4+ T lymphocyte counts may more likely reflect the drug's pharmacological mechanism—namely, its inhibition of lymphocyte migration. The authors mentioned twice in the article that the study cannot answer whether discontinuing natalizumab would reduce PML incidence, and that the study was not designed to assess the impact of natalizumab dosing interruption on PML risk. Nevertheless, based on the phenomenon of "increased CD4+ T lymphocyte counts" during natalizumab therapy that you highlighted, we have further enriched the depth of our literature support by introducing CD4+ T lymphocyte functional status/subset count characteristics, thereby better completing the evidence chain.

Regarding rituximab, although it directly acts on B cells, previous studies have found that CD4+ T lymphocytes decrease during rituximab maintenance therapy, and CD4+ T lymphocyte counts show an upward trend when maintenance is discontinued, suggesting a synergistic effect between B/T cells. We have also cited the corresponding literature to support this argument.

In summary, we again thank the reviewer for helping us clarify the appropriate scope of this statement and for prompting us to present this point and its limitations more clearly in the Discussion section.In response to the above comments, we have made revisions in the following section (Discussion, 3. Pathogenic Mechanisms and Risk Features of Key Drugs, second paragraph)、.

1. Focosi D, Tuccori M, Maggi F. Progressive multifocal leukoencephalopathy and anti-CD20 monoclonal antibodies: What do we know after 20 years of rituximab? Rev Med Virol. 2019;29(6):e2077. doi:10.1002/rmv.2077. PMID:31369199.

2. Berkovich R, Togasaki DM, Cen SY, Steinman L. CD4 cell response to interval therapy with natalizumab. Ann Clin Transl Neurol. 2015;2(5):570-574. doi:10.1002/acn3.190. PMID:26000328.

3. Haghikia A, Perrech M, Pula B, Ruhrmann S, Potthoff A, Brockmeyer NH, et al. Functional energetics of CD4+ cellular immunity in monoclonal antibody-associated progressive multifocal leukoencephalopathy in autoimmune disorders. PLoS One. 2011;6(4):e18506. doi:10.1371/journal.pone.0018506. PMID:21533133.

4. Dubois E, Ruschil C, Bischof F. Low frequencies of central memory CD4 T cells in progressive multifocal leukoencephalopathy. Neurol Neuroimmunol Neuroinflamm. 2015;2(6):e177. doi:10.1212/NXI.0000000000000177. PMID:26568972.

5. Lavielle M, Mulleman D, Goupille P, et al. Repeated decrease of CD4+ T-cell counts in patients with rheumatoid arthritis over multiple cycles of rituximab treatment. Arthritis Res Ther. 2016;18(1):253.

6. Piantoni S, Scarsi M, Tincani A, Airò P. Circulating CD4+ T-cell number decreases in rheumatoid patients with clinical response to rituximab. Rheumatol Int. 2015;35(9):1571-1573.

7. Yutaka T, Ito S, Ohigashi H, et al. Sustained CD4 and CD8 lymphopenia after rituximab maintenance therapy following bendamustine and rituximab combination therapy for lymphoma. Leuk Lymphoma. 2015;56(11):3216-3218.

Response to Reviewer #2

Dear Reviewer,

Thank you very much for your careful re-evaluation of our manuscript and for your constructive suggestions. We have revised the manuscript accordingly. Point-by-point responses are provided below.

Comment 1) Figure 2 Legends: The legends for Figure 2 are incorrect (2c vs 2d).

Response: Thank you for pointing this out. We corrected the mismatch between the Figure 2c and Figure 2d legends. The revised legend now accurately describes Figure 2c as geographic distribution reporting and Figure 2d as reporter/professional category reporting, and we verified that all in-text citations to Figure 2 subpanels are consistent. (Figure 2 legend; Results, Characteristics of Reported PML Reports in FAERS)

Comment 2) Missing Citations: The second paragraph of the Discussion, Section 3 (lines 323–330), requires a reference citation.

Response: Thank you for noting this. We added supporting citations to the mechanistic statements in this paragraph, including references [1] and [12] for the link between immunosuppression and PML/JCV reactivation, and a review by Cain & Cidlowski [46] for the immunoregulatory mechanisms of glucocorticoids. (Discussion, 3. Pathogenic Mechanisms and Risk Features of Key Drugs)

1.Tan CS, Koralnik IJ. Progressive multifocal leukoencephalopathy and other disorders caused by JC virus: clinical features and pathogenesis. Lancet Neurol. 2010;9(4):425-437. doi:10.1016/S1474-4422(10)70040-5. PMID:20298966.

12.Cortese I, Reich DS, Nath A. Progressive multifocal leukoencephalopathy and the spectrum of JC virus-related disease. Nat Rev Neurol. 2021;17(1):37-51. doi:10.1038/s41582-020-00427-y. PMID:33219338.

46.Cain DW, Cidlowski JA. Immune regulation by glucocorticoids. Nat Rev Immunol. 2017;17(4):233-247. doi:10.1038/nri.2017.1. PMID:28192415.

Comment 3) Punctuation Review: The third paragraph of the Conclusion section and the Limitations section need punctuation corrections.

Response: Thank you. We proofread the Conclusion and Limitations sections and corrected punctuation/typographical issues (e.g., missing spaces after periods, redundant punctuation, and sentence formatting). We also improved sentence readability in the Limitations section without changing the meaning.

---

## [Decision Letter · Decision Letter 2]

13 Jan 2026

Disproportionality Analysis of Drug-Associated Progressive Multifocal Leukoencephalopathy Using Spontaneous Reports: A 20-Year Signal Detection Study Based on the FAERS Database

PONE-D-25-44746R2

Dear Dr. shangguan,

We’re pleased to inform you that your manuscript has been judged scientifically suitable for publication and will be formally accepted for publication once it meets all outstanding technical requirements.

Kind regards,

Mehmet Baysal

Academic Editor

PLOS One

Additional Editor Comments (optional):

Reviewers' comments:

Reviewer's Responses to Questions

**Comments to the Author**

Reviewer #1: All comments have been addressed

Reviewer #2: All comments have been addressed

2. Is the manuscript technically sound, and do the data support the conclusions?

Reviewer #1: Yes

Reviewer #2: Yes

3. Has the statistical analysis been performed appropriately and rigorously?

Reviewer #1: Yes

Reviewer #2: Yes

4. Have the authors made all data underlying the findings in their manuscript fully available?

Reviewer #1: Yes

Reviewer #2: Yes

5. Is the manuscript presented in an intelligible fashion and written in standard English?

Reviewer #1: Yes

Reviewer #2: Yes

Reviewer #1: The manuscript is much improved from the initial submission, in my view, and should be published.

I have a few editorial comments that are not substantive and do not require another review.

line 15: Replace "Identifying" with "We identified"

line 17: delete were, it should read "four defined as"

line 22: "replace "resulted in: with "were associated with"

line 48: Add "In the context of modern HIV treatment," continuing with "HIV-associated PML..."

line 70: Replace "What this study adds: Using" with " This study builds on prior knowledge using"

line 142: replace "age of reports" with "age of patients"

line 151: replace "rather than" with "may not reflect"

line 298: deleted 'sensitivity"

lines 388-389: delete "through a common pathway of impaired immune surveillance" as this was not described in the discussion.

I suggest that supplemental figures described in the text be included, especially the heat map S1 figure and S2 figure

Reviewer #2: (No Response)

**Do you want your identity to be public for this peer review?** For information about this choice, including consent withdrawal, please see our Privacy Policy

Reviewer #1: No

Reviewer #2: **Yes:** Luisa Arueira Chaves

---

## [Editor Report · Acceptance letter]

PONE-D-25-44746R2

PLOS One

Dear Dr. Shangguan,

I'm pleased to inform you that your manuscript has been deemed suitable for publication in PLOS One. Congratulations! Your manuscript is now being handed over to our production team.

Kind regards,

on behalf of

Dr. Mehmet Baysal

Academic Editor

PLOS One